# Clinical Applications of Liquid Biopsy in Non-Small Cell Lung Cancer Patients: Current Status and Recent Advances in Clinical Practice

**DOI:** 10.3390/jcm10112236

**Published:** 2021-05-21

**Authors:** Shinhee Park, Jae-Cheol Lee, Chang-Min Choi

**Affiliations:** 1Division of Allergy and Respiratory Medicine, Department of Internal Medicine, Soonchunhyang University Bucheon Hospital, Bucheon 14584, Korea; myattic23@naver.com; 2Department of Oncology, Asan Medical Center, University of Ulsan College of Medicine, Seoul 05505, Korea; jclee@amc.seoul.kr; 3Department of Pulmonary and Critical Care Medicine, Asan Medical Center, University of Ulsan College of Medicine, Seoul 05505, Korea

**Keywords:** lung cancer, liquid biopsy, non-small cell lung carcinoma

## Abstract

Recent advances in targeted and immune therapies have enabled tailored treatment strategies for advanced lung cancer. Identifying and understanding the genomic alterations that arise in the course of tumor evolution has become hugely valuable, but tissue biopsies are often insufficient for representing the whole cancer genome due to tumor heterogeneity. A liquid biopsy refers to the isolation and analysis of any tumor-derived material in the blood, and recent studies of this material have mostly focused on cell-free tumor DNA (ctDNA) in plasma. Indeed, liquid biopsy analysis is now expected to expand in utility and scope in clinical practice. In this review, we assess the biology and technical aspects of ctDNA analysis and discuss how it is currently applied in the clinic. Key points: Liquid biopsy is a potentially powerful tool in the era of personalized medicine for guiding targeted therapies in non-small cell lung cancer.

## 1. Introduction

Lung cancers are the most commonly diagnosed tumors and are the leading cause of cancer-related deaths worldwide [1]. A delayed diagnosis and the lack of definitive therapeutic options for advanced-stage disease worsen the lung cancer survival outcomes. The field of cancer genetics has created new possibilities however to better diagnose, treat, and monitor lung tumors [2]. Indeed, lung cancer genetic profiling alongside targeted mutation analysis in non-small cell lung cancer (NSCLC) has expanded the treatment options for advanced lung cancer [3,4]. Knowledge of the molecular profile of each lung tumor is invaluable in the era of personalized medicine, particularly with the evolution of targeted therapies. Oncogenic driver mutations such as epidermal growth factor receptor (*EGFR*), anaplastic lymphoma tyrosine kinase (*ALK*), and ROS proto-oncogene 1 receptor tyrosine kinase (*ROS1*) gene rearrangements provide actionable treatment targets for as well as important prognostic insights [5,6,7].

The diagnosis and genotyping of lung cancer typically rely on tissue biopsy obtained from local sampling. This is a clinically well-validated approach, but is invasive and has some inherent clinical risks. Prior meta-analysis has indicated major complication rates from CT-guided transthoracic lung biopsy of 5.7% and 4.4% for core biopsy and fine needle aspiration, respectively [8]. A rebiopsy is also often required in lung cancer patients due to inadequate sampling. For example, previous trials of *EGFR* tyrosine kinase inhibitors (TKIs), such as INTEREST (Iressa NSCLC Trial Evaluating Response and Survival again Taxotere) and IPASS (Iressa Pan-Asia Study), have reported that sufficient biopsy samples were available in only 42% and 31%, respectively, of the patient subjects to allow a complete molecular diagnosis, and consequently that multiple tissue biopsies were required for a complete tumor characterization [9,10]. However, genetic alterations at the time of diagnosis and their evolution during the treatment course would be hard to follow using conventional biopsy methods, considering their invasive nature.

Improvements in genomic and molecular technologies have yielded the identification of various new materials in the bloodstream [11]. A liquid biopsy refers to the analysis of biologic elements such as circulating cell-free tumor DNA (ctDNA), circulating tumor cells (CTCs), and circulating tumor RNA (ctRNA), isolated from peripheral blood. This is therefore a minimally invasive and readily reproducible methodology. More importantly, it accurately reflects spatial and temporal tumor heterogeneity, thus enabling real-time monitoring of treatment responses and resistance [12]. Liquid biopsy analysis is now recommended by the International Association for the Study of Lung Cancer (IASLC) in cases with insufficient or unobtainable tumor tissue specimens [13]. As sequencing technology advances, novel platforms detecting multiple driver mutations with much smaller samples and greater precision are becoming available.

We here review the clinical applications of liquid biopsy in non-small cell lung cancer patients and discuss the challenges and future directions for this methodology.

## 2. Circulating Cell-Free Tumor DNA

Mandel and Metais first reported the existence of circulating cell-free DNA (cfDNA) in the blood of healthy individuals in 1948 [14]. The origin of cfDNA and the mechanism of its release is still unclear, although inflammation, smoking, pregnancy, exercise, autoimmune disease and heart dysfunction have all been shown to contribute to the presence of cfDNA [15]. In 1977, Leon and colleagues reported that the cfDNA levels were elevated in patients with cancer compared to healthy individuals [16]. Tumor-related genetic alterations were subsequently found in the cfDNA of patients with cancer, including mutations in oncogenes and/or tumor-suppressor genes, microsatellite instability, and epigenetic changes [17,18,19,20,21].

Circulating cell-free tumor DNA (ctDNA) is a component of the cfDNA detected in the bloodstream of cancer patients [22]. Previous studies using plasma from pancreatic cancer patients and healthy controls revealed that ctDNA is highly fragmented, ranging from 130 to 170 base pairs [23]. ctDNA also rapidly disappears from the bloodstream, with a reported average half-life of around 15 min [24]. Considering the presence of intratumor heterogeneity, ctDNA is thus suitable for the real-time monitoring of the disease status, since both tumor-derived and wild-type cfDNA can be released into circulation during tumor progression.

The ctDNA levels are influenced by many factors such as disease burden, anatomic locations, vascularity, etc. [25,26,27]. Bettegowda et al. reported that ctDNA was detectable in >75% of patients with advanced pancreatic, ovarian, colorectal, bladder, gastroesophageal, breast, melanoma, hepatocellular, and head and neck cancers, but in less than 50% of cases of primary brain, renal, prostate, or thyroid cancers [25]. These authors also showed that differences in the fraction of patients with detectable levels of ctDNA correlated with their tumor stage, i.e., 47% of patients with stage I cancers of any type had detectable ctDNA, whereas this fraction was 55%, 69%, and 82% for patients with stage II, III, and IV cancers, respectively. Diehl and colleagues analyzed 162 plasma samples from 18 patients with colorectal cancer undergoing surgery or chemotherapy and reported a significant decrease in the ctDNA levels [27].

### Technologies for ctDNA Analysis and Their Clinical Application

The detection of ctDNA can be challenging because it can represent a very small proportion of the total cfDNA pool, ranging from less than 0.1% to over 10% [28,29]. Standard sequencing methods such as Sanger sequencing or pyrosequencing cannot adequately detect ctDNA in cancer patients unless they have a heavy tumor burden and high levels of these molecules. Recent advances in technology have overcome this barrier however i.e., targeted assays using PCR and untargeted methods using next generation sequencing (NGS) or whole genome/exome sequencing. The pros and cons of ctDNA analysis methods covered below are summarized in Table 1.

Targeted assays can detect known mutations, such as those in the *EGFR*, Kirsten rat sarcoma viral oncogene homolog (*KRAS*), or B-Raf proto-oncogene, serine/threonine kinase (*BRAF*) genes [30,31,32]. The emergence of *EGFR* TKI-resistant clones such as *T790M* can also be detected [33]. Two ctDNA assays have been approved for clinical use by the United States Food and Drug Administration (FDA) and the European Medicines Agency for patients with NSCLC who are not eligible for a tissue biopsy or have an acquired resistance to *EGFR* TKIs. These are the Cobas *EGFR* Mutation test v.2 CE-IVD (Roche, Basel, Switzerland) and the Therascreen mutation kits (Qiagen, Hilden, Germany). Other highly sensitive targeted assays are also available such as BEAMing (beads, emulsion, amplification and magnetics) and digital-droplet PCR (ddPCR) [30,34]. However, these assays can miss alterations that fall outside of predefined mutations such as fusion genes.

NGS is a high-throughput sequencing method that can simultaneously interrogate variable areas of the genome and detect somatic mutations. NGS-based assays such as CAPP-Seq are designed to detect multiple mutation types, including insertions/deletions or gene rearrangements, and copy number alterations. Hybrid capture NGS is a well-established method [35,36] that uses predetermined DNA sequences captured by hybridization to biotinylated probes bound to beads. This method does not involve prior amplification, however, and the consequential low input of DNA increases the risk of false positives and sequencing errors. Amplicon-based NGS is designed to amplify predefined sequences of specific genes by PCR [37]. This method can thus be useful when a low DNA input is expected [38,39]. Recently, two NGS-based liquid biopsy tests, the Guardant 360 CDx assay (TherapySelect, Heidelberg, Germany) and FoundationOne Liquid CDx (Foundation Medicine Inc., Cambridge, MA, USA), were approved by the US FDA for the genomic profiling of lung cancer patients [40,41].

## 3. Clinical Applications of Liquid Biopsy in Lung Cancer

### 3.1. Initial Genotyping and Drug Resistance Monitoring in Advanced NSCLC

Recent advances in molecular targeted therapies, such as the use of *EGFR* TKIs and *ALK* TKIs, have significantly improved the survival outcomes for patients with advanced NSCLC. ctDNA assays can be used to for initial genotyping of these lung tumors and can monitor the impact of genotype-directed therapies [10,42,43]. However, acquired resistance to targeted inhibitors during the treatment course is nearly inevitable [44]. By assaying ctDNA, a variety of predefined mutations can be detected and monitored easily.

Currently, *EGFR* mutations are the most commonly tested, and the only variations to be routinely assessed by plasma-based liquid biopsy in clinical practice for NSCLC patients. The specificity of these tests is sufficient to guide the choice of *EGFR*-TKIs based on the test result however the sensitivity is less than satisfactory. A prior meta-analysis of 27 selected studies which included nearly 4000 patients with NSCLC revealed a pooled sensitivity of 60% and specificity of 94% for the detection of *EGFR* mutations in the plasma or serum [45]. In the detection of the *EGFR T790M* mutation, low rates of concordance between the results of a tissue and plasma ctDNA assay were observed in a recent phase III trial of osimertinib [46]. However, this discrepancy can be overcome by using methods with greater analytical sensitivity to obtain higher concordance rates [47]. In several prior retrospective analyses, despite a limited concordance, the response rates for patients who were positive for *EGFR* mutations in the plasma were similar to those in whom *EGFR* mutations were revealed in a conventional tissue biopsy [33,46,48,49]. Oxnard et al. reported that the outcomes of osimertinib therapy in NSCLC patients found to harbor the *EGFR T790M* mutation in a plasma ctDNA test were equivalent to patients classified as positive in a tissue-based assay [33]. Remon and colleagues also reported, in their prior prospective analysis, that osimertinib achieved a 62.5% response rate in NSCLC cases with the *EGFR T790M* mutation based on ctDNA analysis, and that this efficacy was comparable to cases with the *EGFR* T790M mutation detected in a tissue biopsy [38].

In addition to the *EGFR* mutational status, resistance to *ALK* TKIs has been successfully evaluated using a plasma cfDNA test in NSCLC patients. Previous mutational analysis of cfDNA by Bordi and colleagues identified novel *ALK* point mutations in five out of 20 NSCLC subjects treated with crizotinib who had shown disease progression [50]. These authors subsequently reported that *ALK* and *KRAS* mutations are associated with acquired resistance to crizotinib in *ALK*-positive NSCLC, suggesting that an *ALK* mutation detected in plasma can be a useful marker for response monitoring [51]. Another study was subsequently conducted by Dagogo-Jack et al. to establish the clinical utility of longitudinal ctDNA analysis in *ALK*-positive lung cancer [52]. The authors performed longitudinal analyses of the plasma from 22 *ALK*-positive NSCLC patients with acquired resistance to *ALK* TKIs to track the evolution of drug resistance during the treatment course and observed that *ALK* mutations emerged and disappeared during the sequential administration of *ALK* TKIs. This correlation between plasma *ALK* mutations and the response to specific *ALK* TKIs suggested the potential for using a liquid biopsy to guide *ALK* TKI selection. According to recently published results from the LBA81_PR-Phase II/III blood first assay screening trial (BFAST), blood-based NGS found 119 (5.4%) *ALK*-positive disease in 2188 NSCLC patients [53]. The overall response rate of alectinib was 87.4% to 92.0%, supporting the clinical utility of NGS as guidance for the clinical decisions in ALK + NSCLC.

Other oncogenic driver mutations can be detected with ctDNA and used as therapeutic targets. Recently, mesenchymal epithelial transition (*MET*) proto-oncogene emerged as a novel targetable genetic alteration. Drilion et al. studied the efficacy of crizotinib in 69 patients with NSCLC harboring a *MET* 14 exon alteration and showed plasma-derived ctDNA assays can also detect *MET* exon 14 alterations in NSCLC patients [54]. Among 37 patients with *MET*-exon-14-altered tumors and available plasma samples, 18 (49%) were ctDNA positive in plasma samples. Recondo and colleagues tracked genetic alterations during MET TKI therapy using plasma and tissue NGS in 20 patients with advanced *MET* exon 14-mutant NSCLC [55]. In this study, genomic alterations during the treatment course can successfully be traced with tissue and/or plasma NGS; however, the concordance rate between plasma and tissue NGS in 11 paired samples was only 27.3%. The VISION study evaluated the efficacy of tepotinib in 152 NSCLC patients with *MET* exon 14 skipping mutations with serial tissue biopsy and ctDNA analysis [56]. The response rate was similar in tissue and liquid biopsy groups; 48% and 50%, respectively. Among the patients with molecular ctDNA response, tepotinib achieved a disease control rate of 88% according to independent radiologic review. Another emerging actionable target is *RET* gene fusions. Solomon and colleagues performed the analysis of ctDNA and tissue in patients with RET fusion-positive NSCLC who showed disease progression after an initial response to selpercatinib. Interestingly, *RET* solvent front and gatekeeper mutations associated with resistance was detected in ctDNA before disease progression [57].

### 3.2. Use of the Tumor Mutation Burden (TMB) to Guide Immunotherapy

The use of cfDNA analysis has also now been expanded in scope to guide and monitor the response to immunotherapy while other targeted therapies are widely studied and used in advanced lung cancer [58]. The tumor mutation burden (TMB) refers to the number of somatic coding mutations found within the total genome of a tumor [59]. A high TMB can increase the number of various neoantigens expressed on the surface of tumor cells, which can lead to greater tumor immunogenicity. The TMB can be quantified by several NGS-based sequencing technologies, such as whole exome sequencing (WES). WES-based analysis of cfDNA was demonstrated by Koeppel et al. [60]. These authors compared the results of targeted sequencing and WES for tissue biopsy DNA and liquid biopsy samples from 35 patients with various metastatic cancers, including 19 NSCLC cases. Notably, when ctDNA was detectable in the blood samples of these patients, the TMB calculated using cfDNA was found to be positively correlate with that detected from tissue DNA.

It must be noted, however, that the assessment of the TMB using WES can be limited in a clinical setting due to its high cost, long timeframe, and bioinformatic complexity in terms of the large genomic area required to be sequenced. As an alternative approach, targeted gene panels of various genomic size have been developed and validated [59]. Gandara et al. used the FoundationOne CDx assay to measure the blood TMB (bTMB) in samples from two large randomized trials, POPLAR and OAK [61]. These researchers found that the bTMB correlated with the tissue TMB (tTMB) and that the beneficial effects of the anti-PD-L1 antibody, atezolizumab, in patients with metastatic NSCLC were positively correlated with a high bTMB, defined as >16 SNVs detected among 394 genes. Using NGS, Yang and colleagues reported an overall 93.6% sensitivity for mutation detection via the bTMB, and a 0.8 Pearson correlation between the tTMB and bTMB, in 56 patients with stage III/IV NSCLC [62]. In these aforementioned studies, the bTMB showed a good correlation with the response to inhibitors of programmed cell death (PD)1 and its ligand (PD-L1) such as atezolizumab.

The response to immunotherapy can be monitored by changes in the ctDNA level during treatment with immune checkpoint inhibitors (ICIs). Guibert et al. reported that changes in the kinetics of ctDNA from serial plasma samples in advanced NSCLC patients could identify patients who were likely to benefit from anti-PD-1 therapy [63]. Anagnostou and colleagues assessed the ctDNA dynamics in 24 metastatic patients with NSCLC who were treated with ICIs and demonstrated that this test can detect recrudescence at an average of 8.7 weeks sooner, and with better predictive power, than a CT scan [64]. Zulato et al. prospectively enrolled advanced NSCLC patients carrying KRAS mutation and followed them with longitudinal liquid biopsy during treatment with chemotherapy (*N* = 39) and ICIs (*N* = 34) [65]. In this study, the sensitivity of plasma KRAS mutation detection at baseline was 48.3%, and longitudinal plasma KRAS mutational burden was correlated with clinical outcome. Bratman and colleagues conducted ctDNA assays in 316 serial plasma samples obtained at baseline and every three cycles from 94 patients treated with ICIs [66]. Baseline ctDNA level correlated with progression-free survival, overall survival, and clinical response. These studies show that ctDNA assay has the potential for monitoring treatment response in patients treated with ICIs.

Despite promising data from several clinical trials in immuno-oncology, there are some challenges to overcome before TMB can be used in clinical practice: further evaluation of tissue-plasma concordance should be carried out; different methods measuring TMB, e.g., WES and various kinds of targeted NGS, need to be standardized; and a definitive cut-off point for TMB should be agreed upon [67].

### 3.3. Screening for Residual Disease and Predicting Prognosis

Minimal residual disease (MRD) is a concept frequently used with hematologic disorders, which refers to the presence of any tumor-derived material in the blood after curative treatment. The clinical significance of MRD, however, is less well verified for solid tumors. MRD detection in lung cancer using ctDNA has been described and involves measurement of the differences between the presurgical and postsurgical ctDNA levels. A previous prospective study of 41 NSCLC patients measured somatic driver mutations in ctDNA from blood samples that were collected within 10 days before and then after surgical resection [68]. In this study, the average plasma ctDNA mutation frequency was reduced from 8.88% to 0.28% after surgery and 91.7% of the identified plasma ctDNA mutations had a decrease in mutation frequency after surgical resection. Another study reported similar results from 76 stage I to III NSCLC patients who had undergone curative-intent surgical resection in which the average post-operative ctDNA mutation frequency was reduced from 7.94% to 0.28% (*p* < 0.001) [69].

Further studies have shown that ctDNA-based MRD detection can predict the prognosis following a definitive treatment. Newman et al. introduced a highly sensitive ctDNA sequencing method for NSCLC referred to as Cancer Personalized Profiling by deep Sequencing (CAPP-Seq) [70]. In that study, plasma ctDNA was detected in 94% of the patients with a recurrence, indicating that the CAPP-Seq method was effective for MRD detection after definitive therapy in lung cancer. The serial ctDNA follow up data accumulated during the TRACERx trial have provided another example of the prognostic value of ctDNA [71]. In that trial, the patients were followed longitudinally using ctDNA profiling, and the results suggested that the ctDNA status can identify an increased risk of lung cancer recurrence after surgery. Interestingly, the reported pretreatment ctDNA detection rate in this study was only 19% in patients with lung adenocarcinoma in contrast to 97% in squamous cell carcinoma cases. There was, however, no clear explanation for this discrepancy. Chaudhuri et al. conducted a cohort study using CAPP-seq to detect MRD in NSCLC patients after curative-intent therapies [72]. ctDNA was detected in the plasma of more than 50% of the patients in that study who had recurrence later on, and in 72% of the cases showing radiologic progression.

### 3.4. Potential Role of Liquid Biopsy as a Lung Cancer Screening Tool

Since the cancer stage at the time of diagnosis is an important prognostic indicator in lung cancer, proper screening for this disease is crucial to improving clinical outcomes for affected patients [73]. Lung cancer screening is challenging, however, and ctDNA assays may be helpful in terms of specificity in high-risk patients. Two prior nationwide studies conducted in the US and the Netherlands, the NLST and NELSON trials, involving lung cancer screening using computed tomography (CT) in high-risk populations, reported a more than 20% reduction in lung cancer mortality after a 10-year follow-up period [74,75]. However, CT scans require radiation exposure and generate a lot of false-positive results. False-positive findings are problematic, as they create unnecessary follow-ups and invasive procedures such as percutaneous biopsy or surgery. The high specificity of ctDNA analysis may be of great benefit in this regard.

The low sensitivity of ctDNA, particularly for the early-stage detection of lung tumors, is a notable limitation of this methodology as a screening approach. In a previous prospective study of 284 consecutive patients with stage I and II NSCLC, the detection sensitivity cfDNA using ARMS-PCR was only 14.2%, in contrast to a 91.7% specificity [76]. However, even the most sensitive available methods, such as CAPP-seq, which can detect 100% of stage II-III lung cancers, can only detect 50% of stage I lung cancers [70]. Another assay known as CancerSEEK, which combines the genomic analysis of 16 genes on ctDNA and 8 protein biomarkers, was reported to detect 70% of stage I–III cancers of eight types (lung, colorectal, ovary, breast, liver, pancreas, esophagus and stomach) [77]. However, the sensitivity of this method for stage I disease was also low (43%), especially in the case of lung cancer.

Improvements are clearly needed therefore in the sensitivities of the currently available assays for cancer detection, particularly for early-stage disease. A wider coverage of these assays which captures all NSCLCs is also needed before liquid biopsy screening methodologies can be implemented for lung cancer in real world settings.

## 4. Conclusions

The clinical utility of liquid biopsy analysis is now expanding to the diagnosis and treatment of NSCLC. The minimally invasive and readily reproducible nature of this methodology makes it highly suitable for the dynamic evaluation of patients and its most widely used application at present is targeted gene sequencing. Indeed, liquid biopsy kits for *EGFR* and *ALK* mutations have already been approved for clinical use. Liquid biopsy testing is also likely to play an important future role in treatment decisions and resistance monitoring for targeted cancer therapies, and advances in this field could bring significant changes to current clinical practice. In addition, liquid biopsy assays can be used to assess blood TMB which is predictive of the responses to ICIs, in addition to detecting MRD which has prognostic implications and guides adjuvant therapy choices. Several clinical trials are currently active, trying to define clinical relevance and utility of liquid biopsy in advanced lung cancer (Table 2).

Despite the variety of potential clinical applications of a liquid biopsy, these assays have several limitations at present such as a low sensitivity for early-stage disease due to DNA concentration threshold issues, processing capacity of target mutation numbers, and a high rate of false-positives. However, with the continuing progress in sequencing technology, liquid biopsy testing has already begun to change the treatment algorithms for NSCLC. It is still uncertain how much a liquid biopsy can support or even replace conventional tissue biopsy analysis, and more clinical evidence is needed from large-scale prospective trials to address this.

## Figures and Tables

**Table 1 jcm-10-02236-t001:** Comparison of ctDNA analysis methods.

	Coverage	Advantage	Disadvantage
Targeted Assays	Predefined genetic mutations	Highly specific and sensitive Quantatitive	Can miss other alterations or resistance
NGS Whole genome/exome sequencing	Sequencing of full genome/exome	Identification of new targets Investigation of the mechanism of resistance	Low sensitivity Risk of false positive Complex bioinformatics
NGS Panels	Capture or PCR of predetermined region (gene/exon), then sequencing	Simutaneous detection of known and unknown mutations Low cost Less complex bioinformatics	Better but unsatisfactory specificity Imperfect plasma-tissue concordance

**Table 2 jcm-10-02236-t002:** Active clinical trials on liquid biopsy in advanced NSCLC.

NCT	Study Type ^†^	No of Patients	Title	Marker	Research Completion
NCT02511288	Obs	900	LIquid BIopsies in Patients Presenting Non-small Cell Lung Cancer (LIBIL)	ctDNA	March 2026
NCT03721120	Ran	286	Evaluation of the Feasibility and Clinical Relevance of Liquid Biopsy in Patients With Suspicious Metastatic Lung Cancer (LIBELULE)	ctDNA	September 2021
NCT03865511	Int	66	MEchanisms of Resistance in EGFR Mutated Nonpretreated Advanced Lung Cancer Receiving OSimErtib (MELROSE)	ctDNA	June 2024
NCT04258137	Int	332	Circulating DNA to Improve Outcome of Oncology PatiEnt. A Randomized Study (COPE)	ctDNA	April 2024
NCT04265534	Ran	120	KEAPSAKE: A Study of Telaglenastat (CB-839) With Standard-of-Care Chemoimmunotherapy in 1L KEAP1/NRF2-Mutated, Nonsquamous NSCLC (KEAPSAKE)	ctDNA	October 2024
NCT04698681	Int	200	NGS Screening Protocol to Detect Mutation of KEAP1 or NRF2/NFE2L2 Genes for the KEAPSAKE (CX-839-014) Trial	ctDNA	June 2022
NCT04703153	Obs	200	LIQUIK: LIQUId Biopsy for Detection of Actionable Genomic BiomarKers in Patients With Advanced Non-small Cell Lung Cancer (NSCLC) (LIQUIK)	cfDNA	November 2023
NCT04790682	Int	300	LIquid Biopsy to prEdict Responses To First-line immunotherapY in Metastatic Non-small Cell LUNG Cancer. (LIBERTYLUNG)	ctDNA	April 2024

**^†^** Int, Interventional; Obs, observational; Ran, Randomized.

## Data Availability

No new data were created or analyzed in this study. Data sharing is not applicable to this article.

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
