# Peer review of "Clinical Applications of Liquid Biopsy in Non-Small Cell Lung Cancer Patients: Current Status and Recent Advances in Clinical Practice"

_jcm, 2021, doi:10.3390/jcm10112236_

Round 1

Reviewer 1 Report

The paper, similar to other reviews already published in the literature, must be improved with some major and minor revisions.

MAJOR REVISIONS

The review is mainly focused on the role of ctDNA in the management of NSCLC patients; experiences conducted in this field with circulating tumor cells (CTCs) and, more recently, with exosomes should also be reported by the authors.

The authors reported only historical literature data, the paper should also include a paragraph and a table dedicated to ongoing clinical trials using liquid biopsy in NSCLC patients.

About technologies for ctDNA detection, pros and cons regarding the different methods for ctDNA analysis should be added.

MINOR REVISIONS

In the text should be reported another example of clinical application of liquid biopsy in NSCLC represented by patients with MET exon 14 skipping mutations treated with tepotinib such as demonstrated by the VISION study recently published in the NEJM.

The authors should specify that the role of TMB in clinical practice is still unclear. As known, pitfalls regarding TMB for its clinical application include cut-off and reproducibility, differences in gene panel platform, sample storage time and turnaround time.

Reviewer 2 Report

The authors presented a well written and comprehensive review on the role of liquid biopsy in NSCLC. The topic is interesting to the lung cancer oncologists community. I recommend some minor revisions to improve quality and render the manuscript updated:

  1. provide gene names in italic
  2. some spell check is required: first line 'lung cancers ARE the most commonly...'; line two ' A delayed diagnosis and THE lack...'; line three ' options for advanced stage disease worseN' ; Knowledge of the molecular profile of each lung tumor is invaluable in THE era'
  3. in the introduction, authors comment on the low percentage of samples with adequate tissue available for molecular diagnosis in previous trials that were including direct gene sequencing. However, I suggest to comment on the different possibilities and diagnostic  accuracy with novel multigene sequencing platforms that increase the possibility to use small samples for complete tumor characterization.
  4. The sentence 'Notably however, genetic alterations at the time of diagnosis and during the treatment course cannot be detected with conventional biopsy methods' seems not to be appropriate: it is not an absolute case, please rephrase in a way that is more as a possibility.
  5. In the paragraph 'Initial genotyping and drug resistance monitoring in advanced NSCLC', please also discuss experiences with other targeted genes, including RET, MET and KRAS, that are recently available (Recondo G, Clin Cancer Res 2020; Solomon B, J Thorac Oncol 2020; Lin J, Ann Oncol 2020; Tanaka N, Cancer Discov 2021)
  6. in the same paragraph, regarding initial genotyping, I suggest to comment also on the B-FAST and VISION trials, that allow targeted treatments for gene mutations identified only by liquid biopsy
  7. in paragraph 3.2 related to monitoring during immunotherapy, I suggest to add references: Bratman Nat Cancer 2020; Zulato BJC 2020
  8. in the introduction, I suggest to add reference to the sentence on temporal and spatial tumor heterogeneity (i.e Passaro ESMO Open 2020; de Bruin Science 2014)

Round 2

Reviewer 1 Report

The authors  improved the paper significantly, in accordance with our comments.